# Exploring the Tumor-Suppressing Potential of PSCA in Pancreatic Ductal Adenocarcinoma

**DOI:** 10.3390/cancers15204917

**Published:** 2023-10-10

**Authors:** Kexin Li, Qingji Huo, Kazumasa Minami, Keisuke Tamari, Kazuhiko Ogawa, Sungsoo Na, Melissa L. Fishel, Bai-Yan Li, Hiroki Yokota

**Affiliations:** 1Department of Pharmacology, School of Pharmacy, Harbin Medical University, Harbin 150081, China; kexinli0104@gmail.com (K.L.); qinghuo@iu.edu (Q.H.); 2Department of Biomedical Engineering, Indiana University-Purdue University Indianapolis, Indianapolis, IN 46202, USA; sungna@iupui.edu; 3Department of Radiation Oncology, Graduate School of Medicine, Osaka University, Suita 565-0871, Osaka, Japan; k_minami@radonc.med.osaka-u.ac.jp (K.M.); tamari@radonc.med.osaka-u.ac.jp (K.T.); kogawa@radonc.med.osaka-u.ac.jp (K.O.); 4Department of Pediatrics, Wells Center for Pediatric Research, Indiana University School of Medicine, Indianapolis, IN 46202, USA; mfishel@iu.edu; 5Department of Pharmacology and Toxicology, Indiana University School of Medicine, Indianapolis, IN 46202, USA; 6Indiana University Simon Comprehensive Cancer Center, Indianapolis, IN 46202, USA; 7Indiana Center for Musculoskeletal Health, Indiana University School of Medicine, Indianapolis, IN 46202, USA

**Keywords:** pancreatic cancer, PSCA, MSLN, lymphocytes, PBMCs

## Abstract

**Simple Summary:**

Pancreatic ductal adenocarcinoma (PDAC) is a tough cancer. Instead of stopping certain genes, we used their power to change tumors. Surprisingly, some proteins that usually help the cancer grow slowed it down when they were outside the cells. We found that one protein, called PSCA, which is usually bad for patients, could help if we placed it extracellularly. We conducted experiments to see how the cancer cells reacted, and we also looked at how a conditioned medium from the blood could affect the cancer. We found that PSCA could make the cancer cells weaker and less able to spread; it works together with other medicines that are used to fight cancer. This discovery could lead to new ways to treat cancer by using the same protein that was causing the issue. Overall, our study shows that some things that have a negative impact could be used to help fight this tough cancer.

**Abstract:**

Pancreatic ductal adenocarcinoma (PDAC) is an aggressive cancer with low survival rates. We explored an innovative therapeutic approach by leveraging prognostic oncogenic markers. Instead of inhibiting these marker genes, we harnessed their tumor-modifying potential in the extracellular domain. Surprisingly, many of the proteins highly expressed in PDAC, which is linked to poor survival, exhibited tumor-suppressing qualities in the extracellular environment. For instance, prostate stem cell antigens (PSCA), associated with reduced survival, acted as tumor suppressors when introduced extracellularly. We performed in vitro assays to assess the proliferation and migration and evaluated the tumor-modifying capacity of extracellular factors from peripheral blood mononuclear cells (PBMCs) in PDAC tissues. Molecular docking analysis, immunoprecipitation, Western blotting, and RNA interference were employed to study the regulatory mechanism. Extracellular PSCA recombinant protein notably curtailed the viability, motility, and transwell invasion of PDAC cells. Its anti-PDAC effects were partially mediated by Mesothelin (MSLN), another highly expressed tumor-associated antigen in PDAC. The anti-tumor effects of extracellular PSCA complemented those of chemotherapeutic agents like Irinotecan, 5-Fluorouracil, and Oxaliplatin. PSCA expression increased in a conditioned medium derived from PBMCs and T lymphocytes. This study unveils the paradoxical anti-PDAC potential of PSCA, hinting at the dual roles of oncoproteins like PSCA in PDAC suppression.

## 1. Introduction

Pancreatic ductal adenocarcinoma (PDAC) is a highly aggressive malignancy known for its rapid metastasis to organs such as the liver, lung, bone, peritoneum, and others [1,2]. While surgical intervention remains a primary therapeutic option for non-metastatic cases, its effectiveness diminishes in the context of metastatic or recurrent PDAC [3]. Despite the use of chemotherapy regimens like FOLFIRINOX (comprising Folinic Acid, Fluorouracil, Irinotecan, and Oxaliplatin), gemcitabine combined with nab-paclitaxel, or erlotinib, the improvement in treatment outcomes remains limited [4,5]. Given the persistently low long-term survival rates, the quest for effective therapeutic strategies in PDAC remains an urgent imperative [6]. In the pursuit of novel approaches, induced multipotent stem cells (iPSCs) have garnered attention. These iPSCs are induced using a quartet of oncogenic transcription factors: cMyc, Klf4, Oct4, and Sox2 [7,8,9]. However, a subset of these reprogrammed cells undergo a transition to a senescent state, aptly named oncogene-induced senescence (OIS) cells [10], as many cells fail to attain full pluripotent reprogramming capability [11,12]. Interestingly, these OIS cells, although not pursuing pluripotency, exhibit intriguing anti-tumor properties by actively secreting tumor-suppressing proteins [10]. This enigmatic attribute raises a thought-provoking question: could the activation of oncogenic signaling potentially lead to the development of a tumor-suppressive proteome that holds promise for PDAC treatment?

As a cell-based option for treating advanced cancer, a counterintuitive concept of induced tumor-suppressing cells (iTSCs) has been developed [13]. Paradoxically, the generation of iTSCs required the activation of cell proliferating and oncogenic signaling, analogous to OIS cells. We reported that iTSCs were created through the overexpression of cMyc and Oct4 in iPSCs, β-catenin in Wnt signaling [14], Akt in PI3K signaling [15], and Snail [16] in the epithelial-to-mesenchymal transition [17,18]. In addition to the gene overexpression approach, they were produced by pharmacological agents that activated the Wnt, PI3K/Akt, and cAMP/protein kinase A (PKA) pathways [18,19,20,21]. Thus far, multiple lines of evidence have shown that the systemic and local administration of iTSC-derived conditioned medium (CM) effectively inhibits the growth of mammary tumors, osteosarcoma, tumor-induced osteolysis, and lung invasion in mouse models [17,18,19]. Notably, iTSCs can be generated from tumor and non-tumor cells, including breast cancer cells, osteosarcoma cells, prostate cancer cells, and PDAC cells, as well as mesenchymal stem cells (MSCs), peripheral blood mononuclear cells (PBMCs), and T lymphocytes [17,18,19,20,21].

Through comprehensive mass spectrometry-based proteomics analyses of iTSC-derived CM, it became evident that the enriched pool of tumor-suppressing proteins encompassed entities such as Enolase 1 (ENO1), Moesin (MSN), and Heat shock protein 90 (HSP90AB1) [17,18]. What rendered these findings particularly noteworthy was their dual role—functioning as oncoproteins within tumor cells, while simultaneously manifesting as tumor suppressors within the extracellular milieu. This distinctive duality in the actions of iTSC CM-enriched proteins prompted an intriguing inquiry regarding the extracellular behavior of oncoproteins, which are elevated in the context of PDAC. Given the prevalent dichotomy wherein tumor-suppressing proteins from iTSC-derived CM exhibit opposing behaviors within the intracellular and extracellular realms, the central hypothesis embarked upon was the potential for select oncoproteins within tumor cells to assume a tumor-suppressing role in the extracellular domain. A compelling investigation ensued, wherein, by leveraging data from the cancer genome atlas (TCGA) database [22,23], a roster of ten proteins was meticulously chosen. These candidates, characterized by heightened expression levels within PDAC, were subjected to scrutiny to validate this unconventional hypothesis. In this pursuit, our study provides conclusive evidence that prostate stem cell antigen (PSCA) [24,25], distinguished by its elevated transcript levels in PDAC and its consequential association with significantly diminished survival rates, indeed materialized as an extracellular tumor suppressor. Notably, PSCA has also been identified as a prospective target within the domain of CART-cell immunotherapy for PDAC [26,27].

Within the framework of this study, we ventured into the generation of anti-PDAC secretomes, in which the distinctive participation of PSCA as an atypical tumor-suppressive protein was discerned. These secretomes were extracted from PBMCs, a resource obtainable from both healthy individuals and patients grappling with PDAC. PBMCs have gained prominence as a pivotal source for CART-cell immunotherapy, while the application of secretomes sourced from MSCs has been harnessed in therapeutic endeavors spanning musculoskeletal regeneration and wound healing, as evidenced in [28]. Through the modulation of PKA and AMPK, our study effectively illustrated that PBMC-derived CM is endowed with anti-PDAC attributes, further accentuated by the heightened presence of PSCA.

## 2. Materials and Methods

### 2.1. Cell Culture 

Human pancreatic cancer cells such as PANC1 and AsPC-1 (ATCC, Manassas, VA, USA), as well as PANC10.05, PANC198, and Pa03C (obtained from Dr. A. Maitra at MD Anderson Cancer Center, Houston, TX, USA), were cultured in DMEM (Corning, New York, NY, USA). Human mesenchymal stem cells (MSCs) (#PT-2501, Lonza, Basel, Switzerland) were grown in MSCBM (#PT-3238, Lonza). Jurkat T lymphocytes and PC-3 human prostate cancer cells (ATCC) were cultured in RPMI1640. The culture medium was enriched with 10% fetal bovine serum (FBS) and 1% antibiotics, and cells were incubated at 37 °C in a 5% CO_2_ atmosphere. Cells were treated with 20 µM CW008 (#5495, Tocris Bioscience, Minneapolis, MN, USA) as an activator of PKA signaling [29] or 50 µM of Dorsomorphin (#3093, Tocris Bioscience) as an inhibitor of AMPK signaling [30] for one day. Tumor cells were treated with 2 µg/mL of recombinant PSCA proteins (#MBS2012408; MyBioSource, San Diego, CA, USA), or chemotherapeutic agents, such as Irinotecan (#2688, Tocris Bioscience), 5-Fluorouracil (#3257, Tocris Bioscience), and Oxaliplatin (#2623, Tocris Bioscience).

### 2.2. Preparation of CM

For the in vitro experiments, the collected CM underwent an initial round of centrifugation at 2000 rpm for 10 min. Subsequently, the resulting supernatants were subjected to a secondary centrifugation at 4000 rpm for 10 min, followed by filtration through a 0.22-μm polyethersulfone membrane (Sigma, St. Louis, MO, USA).

### 2.3. Human Peripheral Blood Mononuclear Cells (PBMCs)

The use of human peripheral blood samples adhered to the principles outlined in the Declaration of Helsinki and received official approval from the Ethics Committee of Osaka University (protocol #21344). Eight milliliters of peripheral blood were obtained from ten healthy volunteers, and the collected blood samples were diluted with an equal volume of 0.9% NaCl solution. Following this, 6 milliliters of the diluted sample was mixed with 3 milliliters of a lymphoprep solution (#1114544, Abbott Diagnostics Technologies AS, Norway). Following centrifugation at 800× *g* for 30 min at room temperature, PBMCs were harvested using a Pasteur pipette. The characterization of PBMCs included forward and side scatter gating, which was performed using the FACS Aria II flow cytometer (Becton, Dickinson and Company, Franklin Lakes, NJ, USA). The cultured PBMCs were nurtured in AlyS705 medium (Cell Science & Technology Institute, Inc., Sendai, Japan), and the final protein concentration of the resulting CM was adjusted to 230 µg/mL.

### 2.4. MTT Assay

MTT-based metabolic activity assessment involved the seeding of approximately 2000 cells into 96-well plates (#3585, Corning, Glendale, AZ, USA). The subsequent day saw the introduction of treatment agents—comprising CM, drugs, or recombinant proteins—followed by an incubation period of 37 °C spanning two days. On day 4, cells were subjected to staining with thiazolyl blue tetrazolium bromide (0.5 mg/mL solution, #M5655, Sigma) for 4 h. The next step involved adding a solution consisting of isopropyl alcohol (#A416-4, Thermo Fisher Scientific, Waltham, MA, USA) and hydrochloric acid (#3750-32, Ricca Chemical, Arlington, TX, USA) in a ratio of 60:1. The evaluation of metabolic activities was carried out by gauging the optical density at 562 nm, accomplished using a multi-well spectrophotometer.

### 2.5. Scratch Assay

The assessment of cancer cell two-dimensional migratory tendencies through a wound-healing scratch assay involved the quantification of the scratch area. Initially, approximately 3 × 10^5^ cells were placed for each well of 12-well plates. On day 2, a scratch was generated across the cell layer with the tip of a plastic pipette. After the scratch, the cells underwent a wash with DMEM to eliminate any floating cells. At intervals of 0 h and 24 h, images of the cell-free zones were captured using a light microscope at 40× magnification (Nikon, Tokyo, Japan). The quantification of cell-free zone areas was achieved via Image J 1.52a.

### 2.6. Transwell Invasion Assay

The assessment of tumor cell invasiveness was executed employing a 12-well plate and transwell chambers (#353182, Thermo Fisher Scientific), characterized by an 8-µm pore size. The transwell chambers underwent pre-coating with 300 µL of Matrigel (100 µg/mL) that was subsequently polymerized and air-dried for an overnight period. Each chamber was supplemented with 500 µL of serum-free medium, and following a 1-h incubation, the chamber underwent three rinses with serum-free medium. Subsequently, roughly 7 × 10^4^ cells suspended in 300 µL of serum-free DMEM were introduced into the upper chamber, with the lower chamber receiving 800 µL of CM. After a 48-h incubation, the cells situated on the upper surface of the membrane were eliminated. To evaluate the extent of cell invasion through the membrane’s lower surface, the cells were fixed and subjected to staining with a mixture of 100% methanol and Crystal Violet (diluted at a ratio of 1:25 in water) for 30 min. After the staining procedure, at least five randomly selected images were captured utilizing an inverted optical microscope (magnification at 100×, Nikon). The quantification of the average number of stained cells, indicative of the invasion capacity, was subsequently carried out.

### 2.7. Live-Cell Imaging

PANC1 cells were grown on the glass-bottomed dishes (MatTek) and transfected with either a fluorescence resonance energy transfer (FRET)-based Src biosensor (obtained from Dr. Y. Wang, University of California, San Diego, CA, USA) [31] or an enhanced green fluorescent protein (EGFP) β-catenin fusion proteins (a gift from E. Schuman, California Institute of Technology, USA) [32] using lipofectamine 2000 (Invitrogen, Waltham, MA, USA). The transfected cells were then incubated for 24–36 h before imaging. A Ti-E fluorescence microscope (Nikon) equipped with an Evolve 512 CCD camera (Photometrics, Tucson, AZ, USA) and a Lambda 10-3 filter changer (Sutter Instruments, Novato, CA, USA) was used for imaging Src activity and β-catenin localization. For FRET-based Src activity measurements, CFP excitation (438/24 center wavelength/bandwidth in nm), CFP emission (483/32), and FRET emission (542/27) filter sets were used. For EFGP intensity-based β-catenin localization, GFP excitation (438/24) and GFP emission (520/35) filter sets were used. During imaging, cells were incubated with live cell imaging solution (Thermo Fisher) to maintain pH 7.4. The ratios of CFP emission to FRET emission were used to visualize the Src activities. To visualize β-catenin localization, the GFP images were background-subtracted and averaged over the nucleus. The FRET images for Src activities were generated using the NIS-elements Version 3.10 software (Nikon), and the GFP images for β-catenin localization were created with the ImageJ 1.52a software (NIH). The images were scaled according to the color bars.

### 2.8. Western Blot Analysis and ELISA 

Cells were lysed with a RIPA lysis buffer (#sc-24948, Santa Cruz Biotechnology, Dallas, TX, USA), including protease inhibitors (#PIA32963, Thermo Fisher Scientific) and phosphatase inhibitors (#2006643, Cal-biochem, Billerica, MA, USA). Proteins were fractionated using 4 to 15% SDS gel (#4561085m Bio-Rad Laboratories, Hercules, CA, USA), and they were transferred to a polyvinylidene difluoride membrane (#IPVH00010, Millipore, Billerica, MA, USA). The membrane was first incubated for 18 h with the primary antibodies, and it was then incubated with secondary antibodies for 45 min (#7074S/7076S, Cell Signaling, Danvers, MA, USA). Antibodies we employed were p-Src, Src, β-catenin, cleaved caspase 3, caspase 3, MSLN (Cell Signaling), and PSCA, K-Ras (Santa Cruz Biotechnology). Of note, β-actin (Sigma, Saint Louis, MO, USA) was used as a control. SuperSignal west femto maximum sensitivity substrate (#PI34096, Thermo Fisher Scientific) was used to detect the protein level. We used a luminescent image analyzer (#LAS-3000, Fuji Film, Tokyo, Japan) to quantify the signal intensities [33]. Western blot analysis and an ELISA kit (#MBS4502065, MyBioSource, San Diego, CA, USA) were employed to evaluate the concentration of PSCA in CM. The original western blot figures can be found in Appendix A.

### 2.9. siRNA Transfection

To achieve the silencing of MSLN, RNA interference was implemented using its dedicated siRNA (#20456; Life Technologies, Carlsbad, CA, USA), using a nonspecific control siRNA as a negative control (Silencer Select #1, Life Technologies). The transfection process was facilitated using lipofectamine RNAiMAX (#13778075, Life Technologies), and the efficacy of the silencing endeavor was evaluated using Western blotting, conducted 24 h post-transfection.

### 2.10. Membrane Protein Extraction 

The extraction of membrane proteins was carried out utilizing the mem-PERTM Plus membrane protein extraction kit (#89842, Thermo Fisher Scientific), adhering to the manufacturer’s provided protocol. To give a concise overview, the procedure involved the addition of a permeabilization buffer to the cell pellet, which was briefly vortexed and subsequently incubated at 4 °C with continuous mixing. Following the centrifugation step, the resulting supernatant was meticulously removed and discarded. Subsequently, the pellet underwent resuspension with the addition of the solubilization buffer. Upon the subsequent centrifugation, the supernatant—encompassing solubilized membrane and membrane-associated proteins—was carefully transferred into a new tube, poised for downstream applications.

### 2.11. Molecular Docking Analysis

To evaluate the interactions between MSLN and PSCA, we employed a ZDOCK program (ver. 2016, Discovery Studio, San Diego, CA, USA). The program generated predictions for all potential binding poses in both translational and rotational dimensions between the ligand and receptor. It then assessed each pose using the ZDOCK score, which is an energy-based scoring system [34]. The three-dimensional structure of MSLN was obtained from the Protein Data Bank [35]. Homology modeling of PSCA was performed using a SWISS-MODEL homology-modeling server [36], utilizing the amino acid sequence obtained from the Universal Protein Knowledgebase [37]. A Fab fragment of MORAb-009, an antibody against MSLN [38], was employed as a positive control for molecular docking to PSCA.

### 2.12. Immunoprecipitation

Immunoprecipitation procedures were performed utilizing an immunoprecipitation starter pack kit (#45-000-369, Cytiva, Marlborough, MA, USA), following the protocol outlined by the manufacturer. Protein samples underwent a preparatory step involving pretreatment with agarose beads conjugated with protein A and rabbit IgG. Subsequently, an overnight immunoprecipitation process was conducted utilizing beads coupled with anti-PSCA antibodies. The immobilized beads were then collected via centrifugation, subjected to three successive washes with PBS, and eventually resuspended, making them ready for Western blotting applications. For the Western blotting analysis, antibodies against MSLN (Cell Signaling) were employed.

### 2.13. Ex Vivo Tissue Assay

The use of human PDAC tissue was granted approval by the Institutional Review Board at Indiana University (#1911155674). We received cancer tissues from the Simon Comprehensive Cancer Center Tissue Procurement Core. We employed a freshly isolated PDAC tissue, weighing approximately 1 g, which was manually fragmented into small pieces measuring 0.5 to 0.8 mm in length using a scalpel. These fragments were cultured in DMEM with 10% FBS and antibiotics for one day. Subsequently, the described conditioned medium (CM) was added and incubated for an additional four days, during which changes in the fragment size were assessed.

### 2.14. Evaluation of Tumor-Suppressing Protein Candidates

In the context of this study, we identified tumor-suppressing protein candidates from the TCGA database. Notably, these candidates exhibit significantly elevated transcript levels in patients diagnosed with PDAC when compared to control groups. This heightened expression is associated with a diminished survival rate. Furthermore, these proteins are classified as either secretory proteins or plasma membrane proteins, indicating their presence within the extracellular space. Ten tumor-suppressing protein candidates were employed, including IGHG1, SFN, PSCA, LAMB3, GPRC5A, LZM, KLK10, LUM, APOL1, and CD74 (#MBS2122530, #MBS2009542, #MBS2012408, #MBS2011064, #MBS2123946, #MBS2009419, #MBS2029351, #MBS205400, #MBS2029622, #MBS205925; MyBioSource). In the MTT assay, each of these recombinant proteins (5 μg/mL) was used in the MTT assay. We evaluated the effects of the application of recombinant proteins on the cellular viability (metabolic activities) of Pa03C cells within a two-day timeframe.

### 2.15. Statistical Analysis

In the context of cell-based experiments, we executed three-to-four distinct independent trials, with the resultant data being presented as mean ± S.D. The assessment of statistical significance was conducted through a one-way analysis of variance (ANOVA). Following this, post hoc comparisons against the control groups were carried out utilizing the Bonferroni correction method, where statistical significance was established at a threshold of *p* < 0.05. In the visual representation of the data, the presence of single and double asterisks is indicative of statistical significance at *p* < 0.05 and *p* < 0.01, respectively. 

## 3. Results

### 3.1. TCGA Database-Based Prediction of PSCA as a Tumor Suppressor

To explore the intriguing possibility of oncoproteins within PDAC cells functioning as extracellular tumor-suppressing agents, a set of ten proteins was chosen using the following selection criteria (Figure 1A). Firstly, these proteins exhibited notably heightened transcript levels in PDAC compared to normal tissue samples, with fold changes exceeding 22 times. This magnitude corresponds to a logarithmic ratio (base 2) of 4.5 or higher. Secondly, the selected proteins were identified as either cell-membrane proteins or secreted proteins, implying their potential presence in the conditioned medium (CM) derived from PDAC cells. Thirdly, the heightened transcript levels of these chosen proteins were associated with diminished patient survival rates (with statistical significance at *p* < 0.05 in a 50% low vs. 50% high comparison) in cases of PDAC and other cancer types, with the exception of LZM and CD74. To assess the effects of these selected proteins, an in vitro assay utilizing the Pa03C PDAC cell line was conducted. Among the ten proteins administered at a concentration of 5 μg/mL for a duration of 48 h, seven proteins demonstrated a significant reduction in MTT-based cell viability (Figure 1B). Noteworthy among these seven proteins, which included IGHG1, SFN, PSCA, GPRC5A, LZM, KLK10, and CD74, was the pronounced inhibitory impact of PSCA. As a result, our subsequent focus predominantly centered on the functional attributes of PSCA. Furthermore, the sensitivity of the PANC1 PDAC cell line to PSCA was also confirmed through the MTT assay, yielding an estimated IC_50_ value of 1.8 μg/mL (Figure 1C).

### 3.2. The Anti-Tumor Effect of PSCA Recombinant Protein

In addition to its impact on the MTT-based cell viability, the administration of recombinant PSCA proteins at a concentration of 2 µg/mL exhibited inhibitory effects on both scratch-based cell migration within 24 h and transwell invasion over 48 h, evident in both the Pa03C and PANC1 PDAC cell lines (Figure 2A–D). Notably, PSCA exerted a downregulating influence on tumorigenic proteins like p-Src and β-catenin, while concurrently upregulating cleaved caspase 3. This pattern of protein modulation pointed towards the activation of the apoptotic pathways (Figure 2E). In addition to its cytotoxic effects, it was observed that extracellular PSCA also exhibited the capacity to downregulate K-Ras, a protein frequently implicated in the dysregulation of PDAC (Figure 2F).

Concurrently, the outcomes from the live-cell imaging, utilizing fluorescently labeled Src biosensor and β-catenin, consistently aligned with the effects observed on the cell viability and tumorigenic protein expression. These results revealed that the extracellular application of PSCA led to the inhibition of Src activity and hindered the translocation of β-catenin to the nucleus (Figure 2G).

### 3.3. Linkage of PSCA with Mesothelin (MSLN)

We have provided evidence for the tumor-suppressive capabilities of extracellular PSCA through various assays assessing cell viability, 2-dimensional motility, and transwell invasion. Notably, the transcript levels of PSCA were significantly elevated in the pancreatic adenocarcinoma tissues, as evidenced by the data from the TCGA database (Figure 3A). Moreover, this heightened expression of PSCA corresponded to diminished survival rates among both pancreatic adenocarcinoma patients (with a *p*-value of 0.0036 for a 25% low vs. 75% high comparison) and patients across all cancer types (with a *p*-value of <10^−3^ for the same comparison) (Figure 3B). The STRING database [39], which is recognized for its presentation of protein-protein association networks, highlighted potential interactions between PSCA and MSLN. In our network diagram, the co-expression of PSCA with two proteins, MSLN and LY6E (lymphocyte antigen 6E), along with their predicted interaction with PSCA, was shown (Figure 3C). Despite our attempts, immunoprecipitation experiments could not confirm a potential interaction between PSCA and LY6E. Consequently, our focus shifted towards mesothelin (MSLN), a crucial cell-surface protein. It is important to note its expression across all five selected PDAC cell lines (Figure 3D). In line with these observations, the TCGA database also reported elevated transcript levels of MSLN in the pancreatic adenocarcinoma tissues. Strikingly similar to PSCA, high transcript levels of MSLN were associated with unfavorable survival outcomes, specifically among pancreatic adenocarcinoma patients (with a *p*-value of 0.002) (Figure 3E).

Correspondingly, employing the protein docking program ZDOCK, projections unveiled the plausible binding of MSLN to PSCA, revealing a spectrum of 21 potential interactions spanning between 1.7 and 2.9 Å (Figure 4A, Appendix A). Notably, when compared to the ZDOCK score of 18.2 achieved by the positive control utilizing a Fab fragment of MORAb-009 as an MSLN antibody, the projected ZDOCK score for the PSCA-MSLN complex stood at 17.5, which was remarkably proximate to the efficacy of the positive antibody control. Subsequently, the immunoprecipitation assay targeting PSCA yielded results wherein MSLN was co-immunoprecipitated with PSCA (Figure 4B). Furthermore, when MSLN was specifically silenced via its corresponding siRNA in PANC1 cells, the suppression of the PSCA-induced inhibition of MTT-based cell viability was observed (Figure 4C,D). Expanding on the MSLN-mediated tumor-suppressive role of PSCA, the introduction of PSCA to these PDAC cell lines led to a reduction in the levels of MSLN (Figure 4E). In summation, the existing dataset lends support to the notion of extracellular PSCA exerting anti-tumor actions, conceivably facilitated through interactions with MSLN.

### 3.4. Additive Anti-Tumor Effects of PSCA with Standard-of-Care Chemotherapy Drugs

To corroborate the potential of PSCA as a therapeutic avenue, our subsequent investigation centered on assessing the compatibility of PSCA application with the existing chemotherapeutic agents. When administering 2 µg/mL of PSCA recombinant protein concurrently with 1 µM Irinotecan, 1 µM 5-Fluorouracil, or 1 µM Oxaliplatin to PANC1 cells, a notably additive effect was witnessed in the reduction in MTT-based cell viability (Figure 5A–C). Echoing this trend, in the scratch-based motility assay, the combination of 2 µg/mL PSCA recombinant protein with 1 µM Irinotecan, 5-Fluorouracil, or Oxaliplatin also resulted in enhanced additive anti-tumor effects (Figure 5D).

### 3.5. Lymphocyte-Derived CM and PSCA

Following the identification of PSCA’s anti-tumor potential, our investigation subsequently delved into the potential role of extracellular PSCA within the CM derived from iTSCs (induced tumor-suppressive cells). To generate tumor-suppressive CM from Jurkat T lymphocytes, we adopted two distinct approaches. One approach involved the activation of PKA signaling using its pharmacological activator, CW008, while the other involved the inhibition of AMPK signaling through the application of its inhibitor, Dorsomorphin. Notably, the outcomes indicated that both the CW008- and Dorsomorphin-treated lymphocyte-derived CMs effectively curtailed the MTT-based cell viability (Figure 6A–C). Remarkably, the level of PSCA was found to be elevated within these two types of CMs (Figure 6D). Further highlighting the significance of these findings, the tumor-suppressive CMs derived from both the CW008 and Dorsomorphin treatments demonstrated a consistent ability to impede the scratch-based motility in PANC1 and Pa03C PDAC cells (Figure 6E,F).

### 3.6. Generation of Tumor-Suppressive CM from PBMCs

To further assess the impact of both PSCA and tumor-suppressive CM, we expanded our investigation to incorporate peripheral blood mononuclear cells (PBMCs) for the generation of iTSC-derived CM. PBMCs offer the advantage of being derived from both healthy individuals and PDAC patients. Just as was observed with Jurkat T lymphocytes, the application of CW008 to PBMCs yielded a tumor-suppressive CM that exhibited stronger efficacy compared to the CM generated from Dorsomorphin-treated PBMCs (Figure 7A). Furthermore, tumor-suppressive CM was successfully generated from the blood samples taken from a PDAC patient. In this scenario, the CM obtained from the CW008-treated PBMCs effectively inhibited the viability of Pa03C cells (Figure 7B). The Western blotting analysis of the CM derived from this PDAC patient indicated significant reductions in p-Src levels, alongside notable increases in cleaved caspase 3 (Figure 7C). To account for potential variability stemming from the sources of PBMCs, we extended our investigation to encompass human peripheral blood samples drawn from ten healthy volunteers. The PBMCs were characterized via forward and side scatter gating utilizing the flow cytometer, which confirmed a predominant lymphocyte population within the mononuclear cell fraction (Figure 7D). In alignment with previous findings, the level of PSCA was observed to be elevated in the CM derived from the CW008-treated PBMCs (Figure 7E). Notably, the application of the CW008-treated PBMC-derived CM led to a reduction in MTT-based viability for PANC1 and AsPC-1 PDAC cells (Figure 7F).

### 3.7. Tumor Selectivity and Ex Vivo Assay with PDAC Tissues

Consistent with the above findings, we have previously shown that the inhibitory action of iTSC-derived tumor-suppressive CM and tumor-suppressive proteins such as Calreticulin, Moesin, etc., is selective to tumor cells, and their inhibitory actions in MTT-based viability are stronger in tumor cells than non-tumor cells. The extracellular PSCA in this study also inhibited the viability of PDAC cells preferentially to that of non-tumor cells such as MSCs (Figure 8A).

To substantiate the viability of protein-based therapeutic interventions utilizing patient-derived blood samples, our final approach entailed an ex vivo assay employing freshly isolated PDAC tissue. In this experiment, we sought to validate the tumor-suppressive potential of PBMC-derived CM using both a blood sample and PDAC tissue sourced from the same patient. Employing this ex vivo model, wherein freshly isolated PDAC tissue was manually fragmented, the application of two distinct tumor-suppressive PBMC-derived CM types—treated with a PKA activator and an AMPK inhibitor—demonstrated a significant reduction in the size of PDAC fragments over a 96-h period (Figure 8B, Appendix A).

## 4. Discussion

Leveraging the wealth of information contained in the TCGA database, this study undertook an innovative approach to unearth dual-function tumor-suppressing proteins. These proteins, operating in the extracellular domain, exhibit an intriguing paradox through their pronounced expression levels within PDAC cells, while concurrently being linked to unfavorable patient survival outcomes. Among the initial pool of ten selected candidates, PSCA emerged as the standout contender, showcasing robust anti-PDAC potential. Although PSCA is recognized as a membrane glycoprotein primarily associated with the prostate [24,40], its upregulation has also been noted in a range of malignancies, including urinary bladder cancer, renal cell carcinoma, ovarian mucinous tumor, and PDAC [41,42,43,44]. The therapeutic efficacy of recombinant human PSCA proteins was distinctly apparent through their ability to curtail MTT-based metabolic activities, hinder two-dimensional motility, and inhibit transwell invasion within PDAC cell lines. Notably, PSCA manifested a significant presence within PBMC-derived tumor-suppressive CM, and its anti-tumor prowess exhibited an additive effect when combined with established chemotherapeutic agents like Fluorouracil, Irinotecan, and Oxaliplatin. Through a collective analysis encompassing molecular docking insights, immunoprecipitation data, and RNA silencing experiments, it became evident that MSLN, a recognized tumor differentiation antigen, played a significant role in mediating the anti-tumor effects of PSCA (Figure 8C).

While PSCA has been suggested as a biomarker and therapeutic target for prostate cancer [25], its high expression level in PDAC makes it a potential target for CAR T-cell immunotherapy [45]. An engineered chimeric antigen receptor against PSCA has been reported to have inhibited the progression of PDAC in a mouse model [46]. MSLN is a glycoprotein and is also elevated in many types of cancer, including ovarian, lung, and PDAC. In addition to PSCA, MSLN is considered a target receptor for CAR T cells [26]. Both PSCA and MSLN are cell surface proteins, and their high transcript level in PDAC leads to poor patient survival [47,48]. The interaction of PSCA and MSLN was predicted by the STRING database, and the results from immunoprecipitation and RNA silencing, as well as molecular docking analysis, supported MSLN-mediated PSCA’s anti-PDAC action. Upon analyzing the network diagram, it is possible that MSLN may not be the exclusive target. As such, our forthcoming research endeavors will encompass the exploration of other potential candidates.

This study presented an unconventional procedure of identifying PSCA as a novel tumor-suppressing protein from a list of oncoproteins that are highly expressed in PDAC. An intriguing question is whether this paradoxical approach can be applied to predict anti-tumor proteins for other types of cancer. In osteosarcoma studies, it is reported that the transcript levels of ENO1, MSN, and HSP90AB1 were higher in osteosarcoma tissues than in normal tissues [49]. These proteins are enriched in iTSC CM, and they act as tumor-suppressing proteins in the extracellular domain [17,18,20]. Thus, the described approach can be tested to select tumor-suppressing protein candidates for other cancer types. Also, this study focused on the target proteins from cell-surface and secreted proteins, but other cytoplasmic or nuclear proteins, such as ENO1 and MSN, may act as extracellular tumor-suppressing proteins. Further analyses are warranted to identify a new set of tumor-suppressing proteins, which are currently known as oncoproteins.

To develop a novel option for treating PDAC, the application of atypical tumor-suppressing proteins, focusing on PSCA, was evaluated in this study. The use of iTSCs and their CM is a possibility, which can be derived from autologous PBMCs, analogous to the development of CAR T-cells in immunotherapy [50]. Another option can be the local and systemic administration of tumor-suppressing protein or a mixture of proteins, similar to secretome-based therapy in regenerative medicine [51]. Instead of employing a whole oncoprotein, we may employ protein engineering and select a core peptide that can exert anti-tumor actions [52]. In this study, we explore the potential of utilizing peripheral blood from PDAC patients to generate autologous CM. The synergistic impact of CM’s anti-tumor properties in conjunction with a conventional chemotherapeutic agent offers promise for clinical trials, complementing the existing drug-based therapies without causing harm. However, the efficacy and safety of the proposed treatment using recombinant proteins and autologous CM should be further evaluated using in vivo assays.

We acknowledge the inherent constraints within the scope of this study. While our research harnessed both PDAC cell lines and primary PDAC tissues, the indispensable need for preclinical evaluations involving animal models remains evident. Our exploration of the anti-tumor attributes within tumor-suppressive CM concentrated primarily on the PSCA-MSLN regulatory axis. PSCA has the potential to engage with various proteins, as suggested by the forecasted protein interaction network. Furthermore, it is crucial to assess the anticancer signaling cascades downstream of MSLN, a factor recognized for its role in fostering cancer cell survival and proliferation through the NFkB signaling pathway [53]. However, it is noteworthy that the landscape of CM encompasses numerous additional proteins, which could potentially exert diverse influences. Furthermore, our investigations revolved around the proteomic dimension, given that the selective removal of constituents, such as exosomes, nucleic acids, and biomolecules, under the 3kD threshold demonstrated minimal impact on the anti-tumor efficacy of CM, as previously elucidated [17,21,49].

## 5. Conclusions

This research highlights the potential of the oncogenes expressed in PDAC to function as unconventional tumor-suppressing factors within the extracellular domain. Both PSCA- and PBMC-derived CMs were observed to effectively hinder the proliferation and migration of PDAC cells. This suggests a promising avenue where the PSCA and CM from iTSCs could present a novel approach for PDAC treatment, either independently or in conjunction with the existing chemotherapy agents.

## Figures and Tables

**Figure 1 cancers-15-04917-f001:**
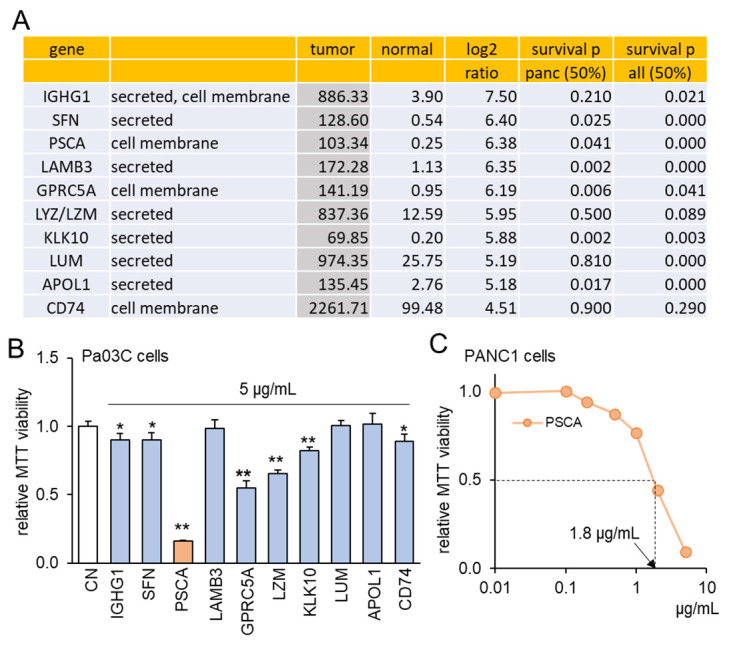
TCGA database-based prediction of prostate stem cell antigen (PSCA) as an extracellular tumor suppressor. CN = control. Of note, * and ** implies *p* < 0.05 and 0.01, respectively. (**A**) In the TCGA database, ten transcripts in the list presented higher levels in PDAC tissues than the normal tissues. The table shows the mean transcript level in the tumor and normal tissues, the log2 fold change (ratio of the level of tumor tissues to the level of normal tissues), and the *p*-values for a comparison of a survival rate between patients with a high transcript level and those with a low level for pancreatic cancer and all types of cancer. (**B**) Reduction in MTT-based proliferation of Pa03C PDAC cells by 10 selected recombinant proteins (5 µg/mL) in 48 h. These tumor-suppressing protein candidates were selected because of their higher expression in PDAC tissues than normal tissues in the TCGA database. (**C**) Dose responses of PSCA in PANC1 PDAC cells, showing IC_50_ of 1.8 μg/mL.

**Figure 2 cancers-15-04917-f002:**
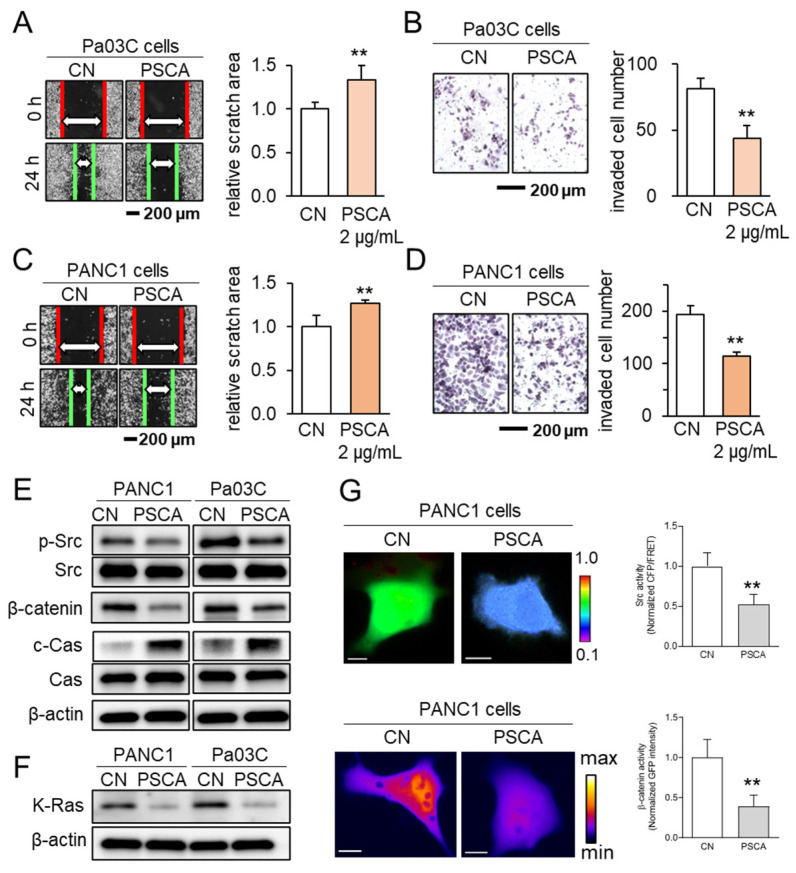
The anti-tumor effects of extracellular PSCA. CN = control, Cas = caspase 3, and c-Cas = cleaved caspase 3. The double asterisks indicate *p* < 0.01. (**A**,**B**) Inhibition in the scratch-based migration (24 h) and transwell invasion (48 h) of Pa03C PDAC cells by 2 µg/mL PSCA recombinant protein. (**C**,**D**) Inhibition in the scratch-based migration (24 h) and transwell invasion (48 h) of PANC1 PDAC cells by 2 µg/mL PSCA recombinant protein. (**E**) Downregulation of p-Src, β-catenin, and elevation of cleaved caspase 3 (c-Cas) by the treatment with 2 µg/mL PSCA in PANC1 and Pa03C PDAC cells. (**F**) Reduction in the level of K-Ras in PANC1 and Pa03C PDAC cells in response to 2 µg/mL PSCA recombinant protein (24 h). (**G**) Suppression of Src activity and β-catenin translocation to the nucleus in PANC1 cells in response to PSCA, in live-cell imaging using a fluorescence resonance energy transfer technique. The color bars indicate the normalized Src activities (top panel) and the concentration of β-catenin (bottom panel). The scale bars indicate 10 μm. *n* > 12 cells.

**Figure 3 cancers-15-04917-f003:**
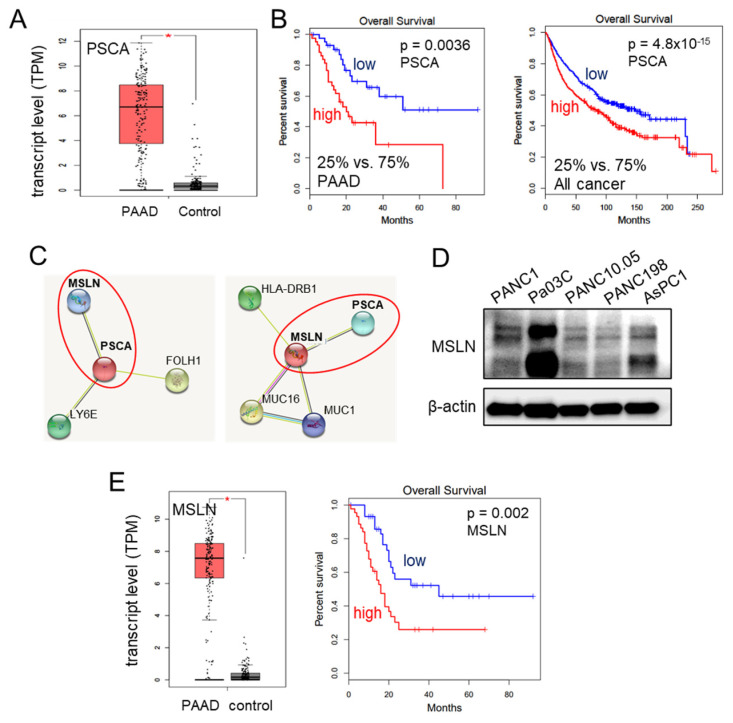
Involvement of MSLN in the action of PSCA. The single asterisk indicates *p* < 0.05. (**A**) Elevation of PSCA transcripts in PAAD (pancreatic adenocarcinoma) tissues in the TCGA database. (**B**) Lowered survival rates of PAAD patients and all cancer types with a higher expression level of PSCA. (**C**) The STRING database highlighted potential interactions between PSCA and MSLN. (**D**) Levels of MSLN in the five PDAC cell lines. (**E**) Elevation of MSLN transcripts in PAAD tissues in the TCGA database, and low survival rate of PAAD patients with a high level of MSLN (*p* = 0.002).

**Figure 4 cancers-15-04917-f004:**
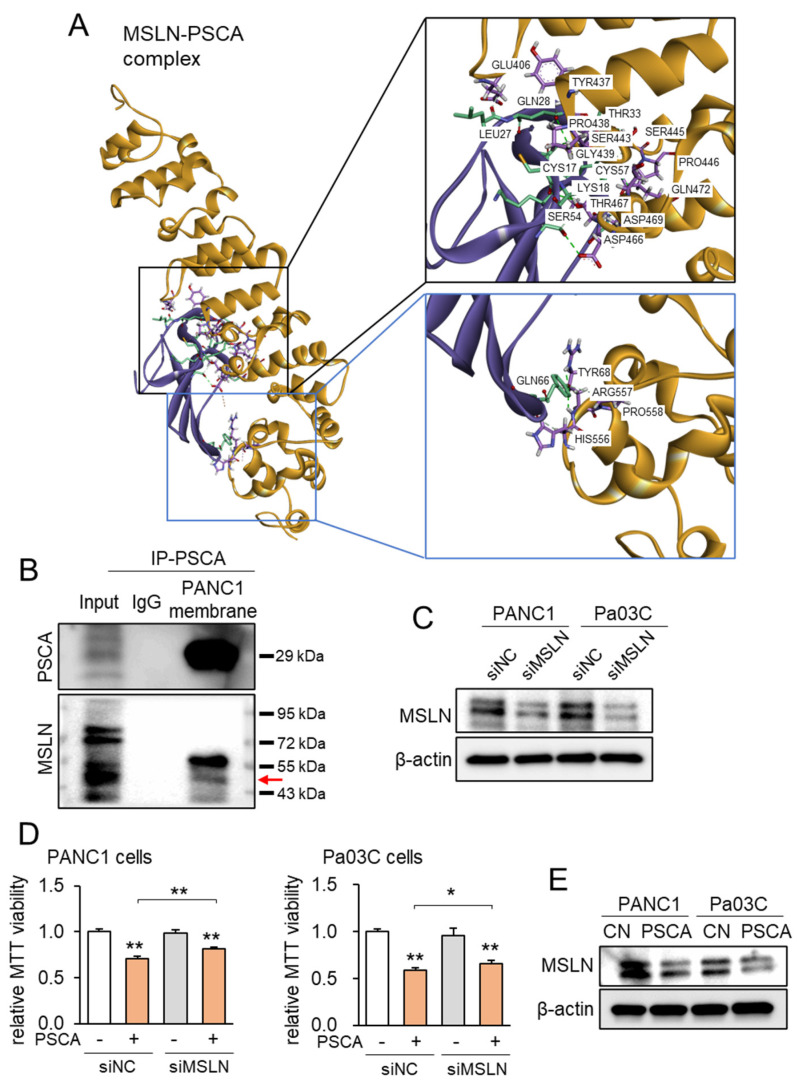
MSLA-PSCA interactions analyzed by molecular docking. CN = control, siNC = nonspecific siRNA, siMSLN = MSLN siRNA. The single and double asterisks indicate *p* < 0.05 and 0.01, respectively. (**A**) MSLN-PSCA complex with the predicted hydrogen bondings. (Left) The three-dimensional configuration includes MSLN (gold) and PSCA (purple). (Right) Detailed molecular interactions of MSLN-PSCA. (**B**) Co-immunoprecipitation of PSCA and MSLN using the PANC1 cell membrane lysate. The horizontal arrow marks the location of MSLN. (**C**) Silencing of MSLN in PANC1 cells and Pa03C cells, respectively. (**D**) Suppression of PSCA-driven decrease in MTT-based viability of PANC1 cells and Pa03C cells, respectively, by RNA silencing of MSLN. (**E**) Reduction in the level of MSLN in PANC1 and Pa03C PDAC cells in response to 2 µg/mL PSCA recombinant protein (24 h).

**Figure 5 cancers-15-04917-f005:**
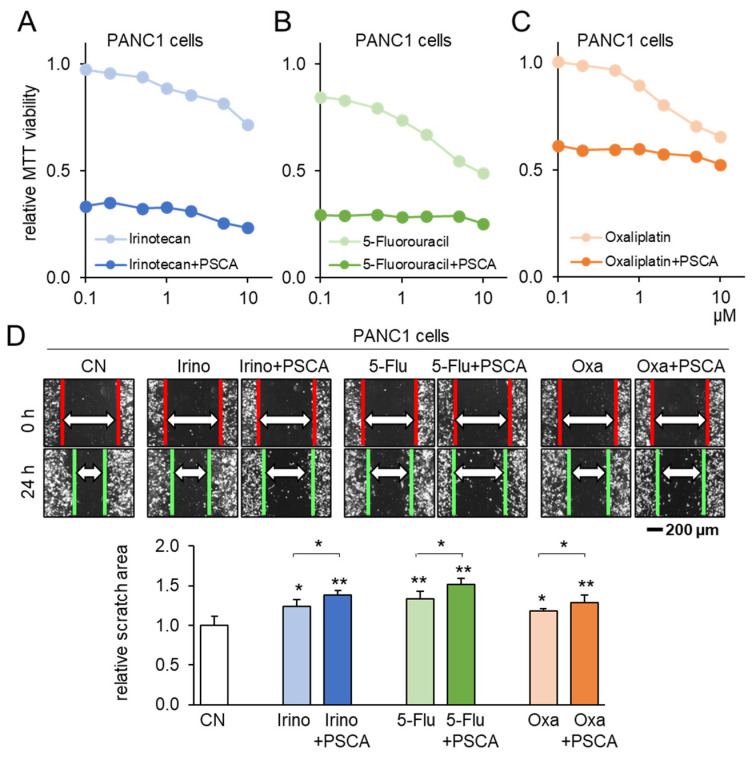
Additive anti-tumor effects of chemotherapeutic drugs and PSCA. CN = control, Irino = Irinotecan, 5-Flu = 5-Fluorouracil, and Oxa = Oxaliplatin. The single and double asterisks indicate *p* < 0.05 and 0.01, respectively. (**A**–**C**) Additive MTT-based anti-tumor effect of 2 µg/mL PSCA recombinant proteins with 1 µM Irinotecan, 5-Fluorouracil, or Oxaliplatin in PANC1 PDAC cells (48 h). (**D**) Additive anti-tumor effects of 2 µg/mL PSCA recombinant proteins with 1 µM Irinotecan, 5-Fluorouracil, and Oxaliplatin in the scratch-based two-dimensional motility assay (24 h).

**Figure 6 cancers-15-04917-f006:**
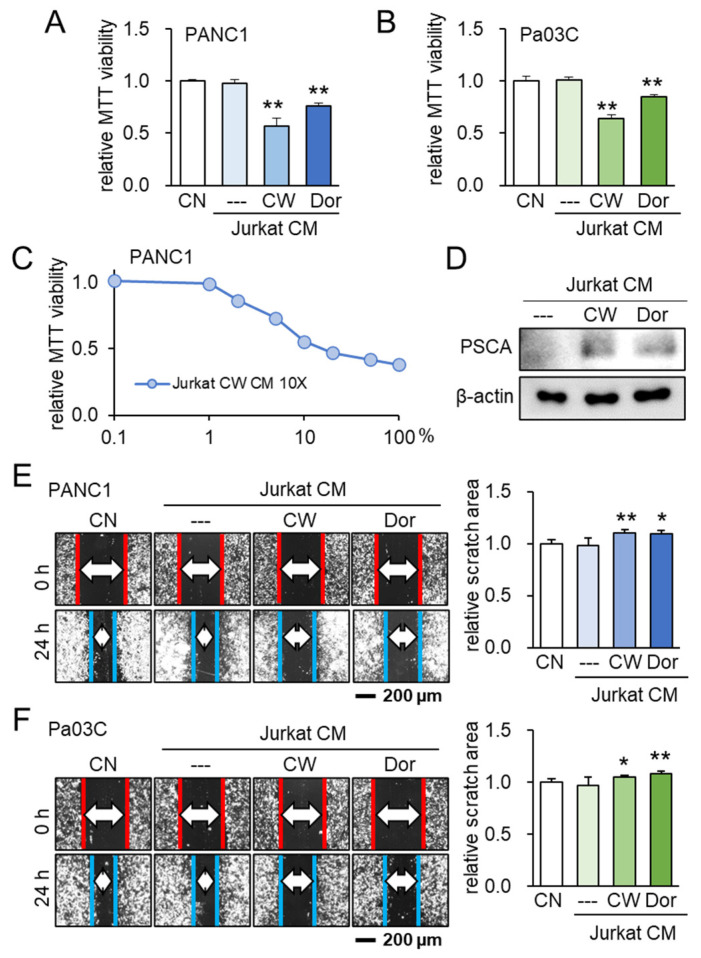
Lymphocyte-derived tumor-suppressive conditioned medium (CM) and PSCA. CN = control, CM = conditioned medium, CW = CW008 as a protein kinase A (PKA) activator, and Dor = Dorsomorphin as an AMP-activated protein kinase (AMPK) inhibitor. The single and double asterisks indicate *p* < 0.05 and 0.01, respectively. (**A**,**B**) Reduction in MTT-based cell viability of PANC1 and Pa03C PDAC cells (48 h) by Jurkat T-lymphocyte-derived CM. Anti-tumor CM was generated by applying 20 μM of CW008 (PKA activator) or 50 μM of Dorsomorphin (AMPK inhibitor) for 24 h. (**C**) Dose responses of MTT-based viability of PANC1 with CW008-treated lymphocyte-derived CM (48 h). CM was condensed 10 times. (**D**) Western blot images, showing the elevated level of PSCA in CW008/Dorsomorphin-treated lymphocyte-derived CM. (**E**,**F**) Decrease in scratch-based motility of PANC1/Pa03C PDAC cells (24 h) by CW008/Dorsomorphin-treated lymphocyte-derived CM.

**Figure 7 cancers-15-04917-f007:**
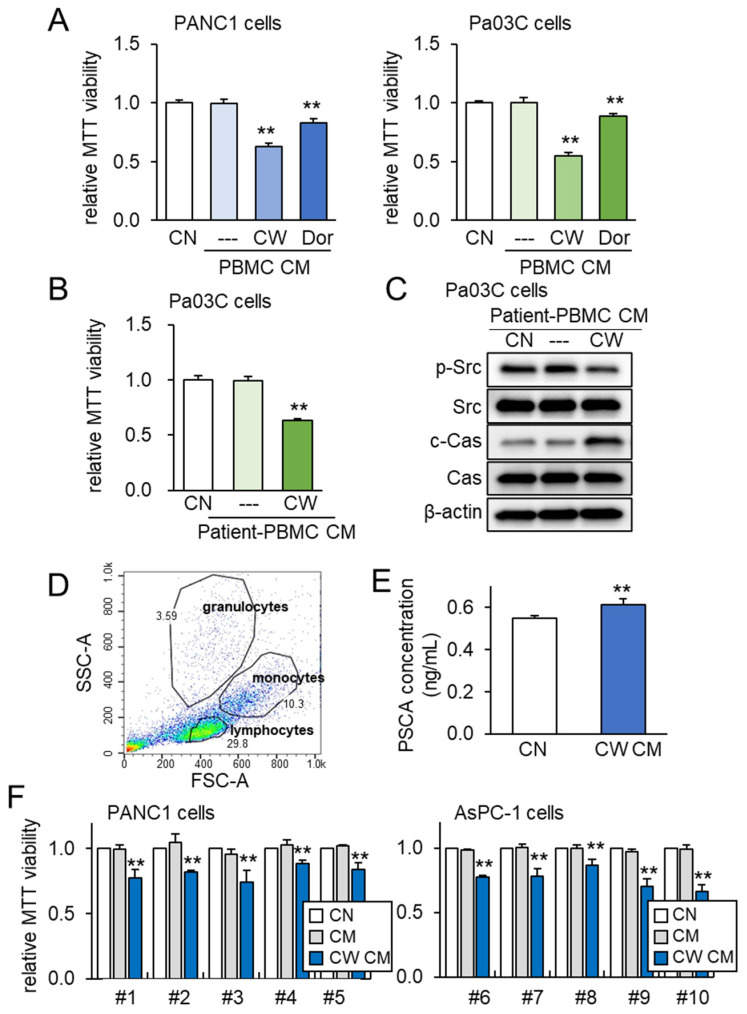
PBMC-derived tumor-suppressive CM and PSCA. The double asterisks indicate *p* < 0.01. CN = control, CM = conditioned medium, PBMC = peripheral blood mononuclear cell, CW = CW008 as a protein kinase A (PKA) activator, and Dor = Dorsomorphin as an AMP-activated protein kinase (AMPK) inhibitor. (**A**) Reduction in the MTT-based viability of PANC1 and Pa03C PDAC cells (48 h) by PBMC CM. The anti-tumor CM was generated by CW008 and Dorsomorphin. (**B**) Reduction in the MTT-based viability of Pa03C PDAC cells (48 h) by CW008-treated PBMC-derived CM. (**C**) Downregulation of p-Src, and elevation of cleaved caspase 3 (c-Cas) in Pa03C PDAC cells (24 h) by the treatment with CW008-treated PBMC-derived CM. (**D**) SSC-A and FSC-A plots in flow cytometric analysis of PBMCs, indicating a major population of lymphocytes together with monocytes and granulocytes. (**E**) Elevation of PSCA in CW008-treated PBMC-derived CM by ELISA. (**F**) Reduction in MTT-based viability of PANC1 and AsPC-1 PDAC cells (48 h) in response to CW008-treated PBMC-derived CM.

**Figure 8 cancers-15-04917-f008:**
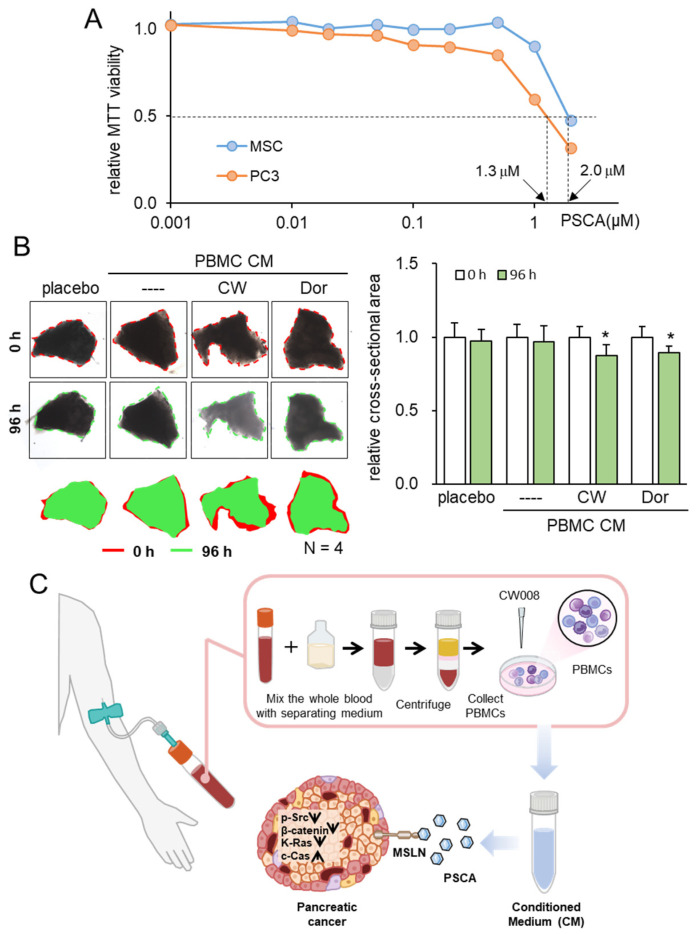
Tumor selectivity and inhibitory effects on ex vivo PDAC tissues. The single asterisk indicates *p* < 0.05. CM = conditioned medium, PBMC = peripheral blood mononuclear cells, CW = CW008 as a protein kinase A (PKA) activator, and Dor = Dorsomorphin as an AMP-activated protein kinase (AMPK) inhibitor. (**A**) Reduction in MTT-based viability of MSCs and PC-3 cells. (**B**) Shrinkage of human PDAC tissue fragments by PBMC CM, which was generated from the autologous peripheral blood. The anti-tumor CM was generated by CW008 and Dorsomorphin. (**C**) Proposed mechanism of the anti-tumor action of PSCA, which was in part mediated by MSLN. The CW008-treated CM derived from PBMCs displayed significant tumor-suppressive properties, leading to the downregulation of p-Src, β-catenin, and K-Ras in PDAC cells. Additionally, it exhibited cytotoxic effects and increased the levels of cleaved caspase 3, an established biomarker of apoptosis. Interestingly, the PSCA protein demonstrated a dual role in this context. While it functioned as an oncoprotein for PDAC cells, it paradoxically exhibited a tumor-suppressive effect when present in the extracellular milieu, acting as an atypical tumor-suppressing protein.

## Data Availability

The data supporting the findings of this study are available within the article, the Appendix A, or from the corresponding author upon reasonable request.

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
