# Peer review of "Exploring the Tumor-Suppressing Potential of PSCA in Pancreatic Ductal Adenocarcinoma"

_cancers, 2023, doi:10.3390/cancers15204917_

Round 1

Reviewer 1 Report

·      Lack of In Vivo Studies: The manuscript heavily relies on in vitro experiments and in silico predictions. There's a significant gap as there are no in vivo studies or animal models to validate the observed effects of PSCA. The absence of in vivo data limits the translational potential of these findings.

·      Mechanism Elaboration: The manuscript lacks an in-depth mechanistic understanding of how extracellular PSCA exerts its tumor-suppressive effects. While it discusses interactions with other proteins, the precise molecular pathways and signaling cascades involved remain unclear.

·      Clinical Application: The manuscript briefly mentions that PSCA has additive anti-tumor effects with standard chemotherapeutic agents. However, it doesn't explore the clinical implications of combining PSCA with chemotherapy in PDAC patients. Providing insights into potential clinical trials or translational applications would enhance the manuscript's impact.

·      Discussion of Limitations: While the manuscript discusses the potential tumor-suppressive role of PSCA, it lacks a comprehensive discussion of the study's limitations. Addressing limitations and potential sources of bias would provide a more balanced interpretation of the results.

·      Clinical Relevance: The manuscript should provide a more explicit discussion on the potential clinical relevance of the findings. How could the discovery of PSCA's tumor-suppressive properties impact the treatment of PDAC patients, and what are the practical implications?

·      Figures and Data Presentation: The manuscript contains multiple figures with limited explanation and data interpretation. Clear and concise figure legends and detailed data interpretation would help readers better understand the results.

In summary, while the manuscript explores an intriguing concept regarding PSCA's potential tumor-suppressive role in PDAC, it has several critical flaws, including the absence of in vivo studies, limited clinical correlation, and insufficient mechanistic insights. Addressing these issues and providing a more comprehensive analysis would strengthen the manuscript's scientific rigor and impact.

Its understandable.

Reviewer 2 Report

Comments on Paper - "Exploring the Tumor-Suppressing Potential of PSCA in Pancreatic Ductal Adenocarcinoma"

The work addresses a significant and clinically relevant topic. However, I have identified several areas where revisions and clarifications could enhance the overall quality:

  • Figure 1: It would be beneficial to include a workflow to show the selection process.
  • Figure 1: You mentioned using the Pa03C PDAC cell line for validation assays. Could you clarify why this choice was made over the PANC1 PDAC cell line? 
  • Line 348: The mention of elevated transcript levels of PSCA in PDAC should be reconsidered as it has already been described in Figure 1a. Please avoid redundant information.
  • Line 353: When discussing the shift in attention to MSLN gene, provide a clear justification for this shift. It is crucial to differentiate why MSLN is chosen for further analysis compared to other genes with altered expression levels.
  • Line 366: Consider moving this section to an earlier position to improve the logical flow.
  • Figure 3: If some parts of Figure 3 are not directly connected to Figures 1 and 3, it is advisable to move them to the Supplementary Figures section.
  • Figure 4B: To enhance the clarity of Figure 4B, consider adding labels to indicate the location of MSLN protein bands on the blots, especially since multiple bands are present.
  • Line 279: The title of this section should be reviewed for appropriateness. Ensure that it accurately reflects the focus of the section.
  • Line 293: Remove the unnecessary "3."

Round 2

Reviewer 2 Report

Thank you for addressing the comments from the first round of review diligently. However, upon careful examination of the latest revision, it has been noted that there is one minor concern that still needs to be corrected.

--Figure 4B: Please add an arrow to accurately indicate the "real" location of MSLN protein bands on the blots. 
